# Experimental Investigation on Hardware and Triggering Effect in Tip-Timing Measurement Uncertainty

**DOI:** 10.3390/s23031129

**Published:** 2023-01-18

**Authors:** Lorenzo Capponi, Tommaso Tocci, Marco Marrazzo, Roberto Marsili, Gianluca Rossi

**Affiliations:** 1Department of Engineering, University of Perugia, Via G. Duranti 93, 06125 Perugia, Italy; 2Baker Hughes, Via F. Matteucci 2, 50127 Florence, Italy

**Keywords:** uncertainty, tip-timing, diagnostics, blade vibration, turbomachinery

## Abstract

Non-destructive testing for structural health monitoring is becoming progressively important for gas turbine manufacturers. As several techniques for diagnostics and condition-based maintenance have been developed over the years, the tip-timing approach is one of the preferred approaches for characterizing the dynamic behavior of turbine blades using non-contact probes. This experimental work investigates the uncertainty of the time-of-arrival of a Blade Tip-Timing measurement system, a fundamental requirement for numerical and aeromechanical modeling validation. The study is applied to both the measurement setup and the data processing procedure of a generic commercial measurement system. The influence of electronic components and signal processing on the tip-timing uncertainty is determined under different operating conditions.

## 1. Introduction

In operating conditions, turbine blades can be subjected to uncontrolled high-amplitude vibrations and thermal-mechanical loads, which are some of the most relevant causes of turbomachinery engine failure [1,2]. To avoid this and to increase the performance, safety and stability of gas turbine engines, several techniques have been developed in the last 50 years for Structural Health Monitoring (SHM) and Non-Destructive Testing (NDT) of turbomachinery engines blades [3,4]. Traditionally, a common system to validate the dynamics of rotating blades is the use of strain gauges [5,6,7]. However, while they have high accuracy and well established signal processing algorithms, strain gauges have limited lifetimes in high-temperature conditions, are intrusive, have complicated means of transmitting data from a rotating system and the provided information is limited to instrumented blades [4]. Unlike vibration, temperature and ultrasound-based techniques [7,8,9,10,11,12,13,14], the Blade Tip-Timing (BTT) approach is one of the most advanced and versatile in situ techniques for axial turbomachinery blade dynamics measurements [15,16,17]. The principle of the BTT approach is based on the measurement of instantaneous blade tip deflection by detecting delays or advances in the arrival time of the blade tip at fixed angular positions [18,19]. Besides eddy current, microwave, magnetoresistive and capacitance sensors [20,21,22,23,24,25,26], optical fibers are the most employed non-intrusive probes in modern BTT measurement systems because of their accuracy and resolution [4,27,28,29,30]. The time-of-arrival is measured by sensors installed on the casing. These generate electrical pulse signals when they interact with the rotating blades. As the time-of-arrival is used for determining the amplitude of deflection of each blade, the measurement uncertainty on the time-of-arrival is fundamental for obtaining reliable information on the dynamics of the system. Today, commercial BTT measurement systems are employed in leading manufacturing industries for power generation (gas and steam) and aircraft engines [27,31]. In fact, the employment of the BTT as an SHM approach can result in cost savings in the validation phase by giving more information concerning the underlying dynamics compared to strain gauge approaches. Moreover, BTT largely extends the lifetime impact of condition-based engine maintenance when validating performance and monitoring turbomachinery operating in power plants and aircraft engines. In fact, the BTT approach may alleviate the growing concern on maintenance costs by replacing scheduled maintenance with as-needed maintenance, saving the cost of unnecessary maintenance on the one hand and preventing unscheduled maintenance on the other hand [32]. For this reason, in the last decade, an extensive effort has been put into BTT system characterization and development [33,34,35,36]. More recently, Tchuisseu et al. [37] proposed an approach for the determination of the optimal probe positioning in a BTT system; Wei et al. [38] developed a BTT data simulator based on a reduced order model, considering mistuning, rotation effect and speed fluctuation; Bornassi et al. [39] built a fitting method based on the 2DOF vibrating model for coupled mode resonances; Mohamed et al. [40] presented a process for validating the finite element (FE) stress and deflection predictions of aero-engine compressor blades under non-rotation conditions, giving the quantified uncertainties associated with FE modelling and the measurement processes. However, there is still no technique capable of producing the quantitative measurement uncertainty required at the engine validation stage. In fact, the literature still lacks a complete theoretical model that defines the uncertainty sources introduced by the signal acquisition system and by the data processing algorithms and their influence on the time-of-arrival estimation.

In this work, the experimental characterization of a generic BTT measurement system based on optical probes is presented. In particular, the uncertainties in the time-of-arrival introduced by two separate sources are investigated: the signal acquisition system (i.e., the electronics and the hardware measurement chain) and the data processing algorithms (i.e., filtering, interpolation, triggering and signal edge detection). In this way, the identification of the most suitable parameters and settings for properly performing tip-timing analysis is obtained in terms of measurement uncertainty.

## 2. Fundamentals of Blade Tip-Timing

A BTT measurement system set-up consists of a specific number of non-contact sensors, installed on a turbomachinery casing [15,41]. Eddy current, microwave, magnetoresistive or capacitance sensors can be used in BTT measurement systems, but, in this research, optical probes with laser source are considered [4,33]. The sensor locations and spacing are defined by the natural frequencies of interest over a given speed range and expected Engine Orders (EOs) and mode shapes. For this purpose, a finite element model of the system can be used for defining the Campbell Diagram [2], where the crossing of EO lines with the resonant frequencies can be identified. The schematic of a BTT measurement system is presented in Figure 1.

The BTT methodology is based on the measurement of the time-of-arrival, that is, the time the blade passes in front of each sensor. The time-of-arrival for a rigid blade, $t_{r}$ (i.e., non-vibrating blade) is defined as [15]:(1)$$t_{r}=\frac{\theta }{\omega },$$
where θ is the angular distance swept by the blade and ω is the angular velocity of the shaft. A vibrating blade will interact with the sensor beam at time intervals different to a purely rigid blade with a constant time gap, Δt, due to interference of vibration on steady blade dynamics. The time-of-arrival for a vibrating blade $t_{v}$ is thus [15]:(2)$$t_{v}=t_{r}\pm \Delta t.$$

The time gap Δt is used by the BTT system to determine the deflection of the blade tip, δ [15]:(3)δ=ωRΔt,
where *R* is the radius at the point measured along the blade. Due to this, the estimation of the uncertainty on the time gap Δt becomes fundamental for obtaining reliable blade deflections [33].

## 3. Hardware Measurement System

As the estimation of arrival time is based on a differential comparison of the signals obtained by the probes, the only potentially relevant uncertainty introduced by the measurement chain (i.e., from the optical probes to the digitized signal) is the relative time shift between the acquisition channels. With the proposed approach, the time shift ϕ between different channels, ideally null, is measured and its contribution to the overall uncertainty can be determined.

### 3.1. Methods and Experiments

The first step of the analysis consists of evaluating the noise floor jitter of the four input/output oscilloscope channels used for the experimental campaign. For this purpose, a square wave is generated at different duty cycle frequencies (i.e., 0.5 kHz, 1 kHz and 2 kHz), representing the blade passing frequencies in real engine measurements and split into the four input channels of the oscilloscope and the maximum jitter between each couple of channels is evaluated (see Figure 2).

In Figure 2b, $h_{i,j}$ is the jitter between the *i*-th and *j*-th channels of the oscilloscope used, i.e., the differential variation in time between channels introduced only by the instrumentation. The expected jitter in an ideal measurement system is null.

Once the jitter of the instrumentation is determined, the analysis is similarly performed on the actual BTT signal acquisition system. In this case, the experimental setup consists of a waveform generator and an oscilloscope for signal acquisition (5 Ms/s sampling frequency, 14-bit resolution) and an LED source, connected to the BTT acquisition system (see Figure 3a). To split the same light signal into the four input channels, three 50:50 fiber couplers are employed.

In Figure 3b, $\phi_{i,j}$ is the time shift between the *i*-th and *j*-th channels introduced both by the oscilloscope and the BTT measurement system.

The generated signals were chosen *a priori* as squared waves with frequencies of 500 Hz, 1 kHz and 2 kHz, to simulate real BTT signals from vibrating blades. The acquired signals are processed as follows: an amplitude normalization is firstly performed to be able to compare more signals simultaneously; a threshold level is set (two different thresholds are used here, 50% and 70% of the signal amplitude on the rising edge); the sampled data points within a 5% range around that threshold are linearly interpolated: in this way, the line of least squares is determined and used to intercept the threshold for defining the arrival time of each peak (see Figure 4). This procedure is repeated for every peak, building a series of arrival times for each input channel.

Under the same operating conditions (i.e., threshold level and signal input frequency), the average values $\bar{\phi }$ and the standard deviation $\sigma_{\phi }$ of the arrival time shift ϕ are calculated considering each pair of channels as:(4)$$\bar{\phi }=\frac{2}{n\;(n-1)}\;\left(\sum_{i,j=1,i<j}^{n}\phi_{i,j}-h_{i,j}\right),$$
(5)$$\sigma_{\phi }=\sqrt{\frac{\sum_{i,j=1,i<j}^{n}{\left(\left(\phi_{i,j}-h_{i,j}\right)-\bar{\phi }\right)}^{2}}{\frac{2}{n\;(n-1)}}},$$
where *n* is the number of channels. In this way, the only effect of the BTT acquisition system is considered and the influence of the experimental apparatus is then neglected. With this approach, the uncertainty introduced by the conditioning hardware is represented by the standard deviation $\sigma_{\phi }$ of the arrival time shift ϕ.

### 3.2. Results

The first analysis showed that the jitter decreases with higher input frequencies, as expected, but most important, it is always constant over the acquisition channels under the same input frequency. This result confirms that the instrumentation itself does not affect the measurement of arrival time, because the difference in time shift between channels due only to the instrumentation is null. However, for the sake of completeness of the proposed methodology, the measured jitter was taken into account as shown in Equations (Equation 4) and (Equation 5).

The analysis focusing on the influence of the acquisition system on the arrival time enables several considerations. The standard deviation $\sigma_{\phi }$ values are shown in Figure 5, normalized by the condition with threshold at 50% and 2 kHz of input frequency.

Figure 5 shows a reduction in the standard deviation $\sigma_{\phi }$ (i.e., uncertainty) as the signal frequency increases, i.e., as the rotor speed increases. This behavior can be explained by the increase in the waveform slope in the triggering region, while the rotational speed increases. In fact, at the 70% threshold level, a lower slope induces higher spatial noise (inversely proportional to the waveform slope) considering same noise floor, then generating higher measurement uncertainty. Moreover, the threshold level of 50% shows an overall significant reduction in measurement uncertainty due to the increase in slope in the triggering region at the same frequency. For this reason, in order to reduce the effect of the noise generated by the acquisition system, the triggering should be set in the area where the signal slope is higher. However, it must be highlighted that the order of magnitude of the standard deviation is on the order of a few cents of a microsecond at the considered frequencies and tends to decrease at higher frequencies. This means that at real turbomachine rotational speed this value can be assumed to be of the order of a few microns or even lower. This value is typically not relevant when deflections of tens of millimeters must be detected, but could become relevant if very small deflections need to be investigated. Then we can conclude that the effect of the conditioning system is typically negligible, but it should be investigated in case there are very small deflections to be measured.

## 4. Data Processing Algorithms

In this Section, the influence of the acquisition parameters on the uncertainty of time-of-arrival is addressed. For this analysis, the most relevant and common data processing techniques are selected and their relevance as sources of uncertainty is determined. The selected parameters are the trigger level and type, detection on the rising/falling edge, filtering process, acquisition voltage range, interpolation methodology and input signal slope.

### 4.1. Methods and Experiments

Trapezoidal waves at different frequency content (20 kHz, 40 kHz and 60 kHz), amplitude and slope are generated and acquired using an oscilloscope. The corresponding sampling frequencies were used at 1 MHz, 1.5 MHz and 2 MHz to guarantee a proper distribution of the interpolated points on the rising and falling edges of the acquired signal. With regard to detections on the rising or falling edges, the blade-sensor interaction generates signals with positive amplitude. The frequency content of the signal was arbitrarily adapted to vary the slope of the acquired signal in the trigger level region. Finally, the rise time of the signals is used for describing the signal slope, since amplitude is kept constant. This is defined as the time it takes for the pulsed wave to rise from 10% to 90% of the maximum amplitude. The parameters used are shown in Table 1.

The trigger level is considered at three different values (i.e., 50%, 65% and 80% of the signal amplitude), applied with two different approaches: a percentage-level approach, which uses six interpolation points to determine arrival time and a fixed-level approach, where eight interpolation points are used. With regards to the data interpolation, three modes are considered: the adjacent mode (i.e., inserting the value of an interpolated point in the value of the most adjacent data point), the linear interpolation and the fourth-order polynomial interpolation. Finally, the signal acquisition voltage range was set to 0.5 V, 1 V and 1.5 V to verify that the different resolution of the acquisition board, caused by the different set of full scales (FS), does not influence the typical FS range of interest. The test parameters used are shown in Table 2, resulting in a total of 324 different combinations of setting parameters (i.e., 3 Rise time × 3 Interpolation types × 3 Full scale levels × 6 Trigger level/type × 2 Edge detection).

For a complete analysis of the tested conditions, the obtained signals were analyzed in two different ways: comparing all of the signals with a reference one (properly selected) and, in order to avoid any dependance on the reference signal properties, comparing each signal with itself.

In the self-based approach, the difference between arrival times from two consecutive probes is determined by:(6)$$\Delta t_{i}=t_{i+1}-t_{i}.$$

This analysis is carried out for all peaks of the signal, obtaining an average value of $\bar{\Delta t}$ characterizing the signal:(7)$$\bar{\Delta t}=\frac{1}{n}\sum_{i=0}^{n}\;\Delta t_{i}.$$

From Equation (Equation 7), a synthetic array of arrival times equidistant from a value equal to Δt are generated and compared with the array of arrival times of the actual signal, determining an array of $\Delta {\left(\bar{\Delta t}\right)}_{i}$:(8)$$\Delta {\left(\bar{\Delta t}\right)}_{i}=\Delta t_{i}-\bar{\Delta t}.$$

Then, the mean $\bar{\Delta \bar{\Delta t}}$ and standard deviation $\sigma_{\Delta \bar{\left(\Delta t\right)}}$ of these values are computed:(9)$$\bar{\Delta \bar{\Delta t}}=\frac{1}{n}\sum_{i=0}^{n}\;\Delta {\bar{\left(\Delta t\right)}}_{i},$$
(10)$$\sigma_{\Delta \bar{\left(\Delta t\right)}}=\sqrt{\frac{\sum_{i=0}^{n}{\left(\Delta {\left(\bar{\Delta t}\right)}_{i}-\bar{\Delta \bar{\Delta t}}\right)}^{2}}{n-1}}.$$

In the reference-based approach, the reference signal is characterized by the following parameters: adjacent interpolation, 1.5 V of full scale, 50% percentage trigger and falling edge detection. In order to compare each signal with the reference one, the difference between the arrival time of the current signal (each signal analyzed has a different set of parameters) and the reference signal is evaluated for each rising edge. This is done in accordance with Equation (Equation 11), where $t_{i}$ is the arrival time of the *i*-th peak of the current signal and $t_{i,r}$ is the arrival time of the *i*-th peak of the reference signal:(11)$$\Delta t_{i}=\left|t_{i}-t_{i,r}\right|.$$

Then, the average value and standard deviation are evaluated as follows:(12)$$\bar{\Delta t}=\frac{1}{n}\sum_{i=0}^{n}\;\Delta t_{i},$$
(13)$$\sigma_{\Delta t}=\sqrt{\frac{\sum_{i=0}^{n}{\left(\Delta t_{i}-\bar{\Delta t}\right)}^{2}}{n-1}}.$$

### 4.2. Results

As the difference in the arrival time uncertainty was found to be negligible with regard to the rising–falling edge detection, the experimental results are shown only for the experiments carried out on the rising edge. Similarly, a comparison of the two approaches (i.e., self- and reference-based) did not show relevant differences in this specific research. This outcome is probably due to the ideal conditions selected as a reference, but in other cases, the two approaches could lead to a different discussion. For this reason, the results of the reference-based approach are given and described. The results are shown in terms of standard deviation of estimated arrival time, normalized by the standard deviation evaluated with the adjacent method and with 50% trigger type.

The first consideration concerns the type of interpolation. Under the same operating conditions, the use of an adjacent interpolation type decreases the uncertainty by an order of magnitude compared to the linear and polynomial fit. This result is clearly visible in Figure 6.

The use of the percentage interpolation type over the fixed type had a large impact on the uncertainty, as shown in Figure 6, Figure 7 and Figure 8. While the trigger level has not shown a significant trend, the usage of the percentage trigger shows a better response in terms of uncertainty, because any signal amplitude variation is always compensated by the percentage approach, differently from the fixed trigger.

### 4.3. Filtering

In order to identify the signal peak necessary to properly perform the percentage trigger analysis, the signal is usually filtered using a dedicated low-pass Infinite Impulse Response (IIR) time domain filter. In this study, the IIR filter assumes three retention values: IIR1 corresponds to no-filtering; the IIR2 value is chosen to have a signal amplitude correspondent to the distance between the signal baseline (representative of absence of blades) and the average of the signal peak (removing noise), then representing the right filtering value to be used; IIR3 is an excessive filtering, underestimating the signal peak. The IIR influence on time-of-arrival is tested at different retention values and trigger percentage values (50%, 65% and 80%) using the percentage level method (six interpolation points used) in adjacent mode. The acquisition voltage range is set to 1.5 V. In this way, a total of 54 combinations of setting parameters are investigated. A summary table with the test parameters used is shown in Table 3.

Finally, a further set of six tests with a fixed trigger at 50%, adjacent interpolation and a full-scale level fixed at 1.5 V is acquired. These data are used to compare them with the set of data with a 50% percentage trigger and an IIR retention value of 0 (IIR1 value). Experiments are performed on both rising and falling edges, with three rise time values: 8.1 ms, 4 ms and 2.7 ms, and are presented in Figure 9 as normalized by the standard deviation evaluated with IIR2 value at 50% trigger type.

Figure 9 shows IIR low-pass filter can significantly improve the results if properly selected: better results are obtained with the IIR2 value, where lower scattering at different trigger levels and lower uncertainty at 50% trigger level (higher signal slope) is observed. Removing filtering while using the percentage trigger exposes it to the noise effect on the amplitude signal estimation, with a consequent change in the trigger position on the waveform and a consequent increase in uncertainty with respect to the IIR2 case. Using excessive triggering, instead, artificially alters the signal amplitude detected bringing uncertainty in the trigger location on the waveform, and again increasing uncertainty in the measurement. This justifies the monotone trend observed changing the triggering value on the IIR2 curve, while the other IIR values suffer scattering in data. Moreover, Figure 9 confirms that the best trigger level to be used is 50%, corresponding to the maximum slope region of the signal. The maximum improvement using the IIR filter was about 0.01 μs (corresponding to a few micron deflections for real operating speed). This low value and the 50% trigger trend are probably due to the nature of the signal used (i.e., synthetically generated signal). With noisier data from real measurements, the result is expected to be more noticeable. For the same reason, the percentage method without a low-pass filter and fixed is almost indifferent in terms of uncertainty.

## 5. Conclusions

This paper analyzes the uncertainty sources on arrival time estimation caused by hardware and the triggering method used in a Blade Tip-Timing measurement. A thorough characterization of the sources of uncertainty enables identification of the most suitable parameters to properly perform tip-timing analysis for diagnostics and numerical modeling validations. Firstly, the uncertainty introduced by the measurement system used for acquiring the raw signals from the probes was investigated. The noise introduced by the system (considering optical probe, then made by laser sources, optical fibers, probes and detectors up to signal digitization) is negligible in cases where the deflections to be measured are of the order of or above some tenth of a milliliter, while it could become relevant if the deflections to be measured are very small (tens of microns). The noise effect becomes larger at lower speed and when the trigger is done in the region of the waveform with reduced slope. On the other hand, the analysis of the data processing algorithms showed that the interpolation approaches leading to lower uncertainties on the arrival time are the adjacent mode. While the full-scale level does not have influence on the uncertainty, the percentage trigger type at a 50% level leads to a lower uncertainty in the range investigated. Finally, even though its effectiveness depends on specific conditions, the low-pass filter generally improves the data, leading to a lower uncertainty in the time-of-arrival, if the filter level selected does not excessively alter the waveform amplitude. As a direction for future developments, it will be essential to determine the source of uncertainty introduced by the algorithms that produce the blade deflection information from the arrival time. 

## Figures and Tables

**Figure 1 sensors-23-01129-f001:**
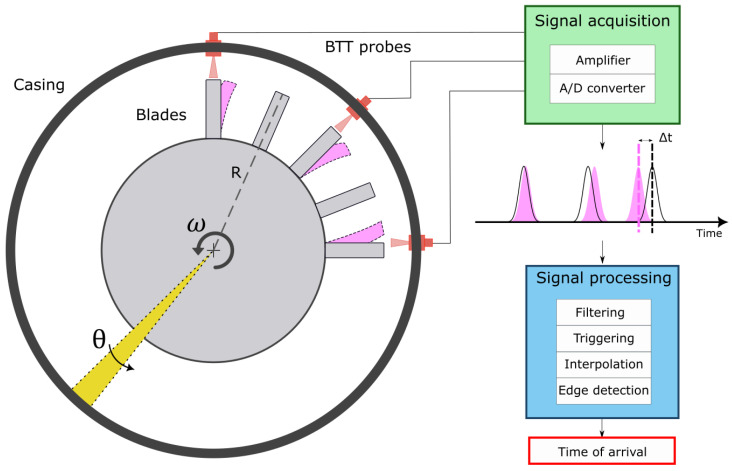
BTT measurement setup.

**Figure 2 sensors-23-01129-f002:**
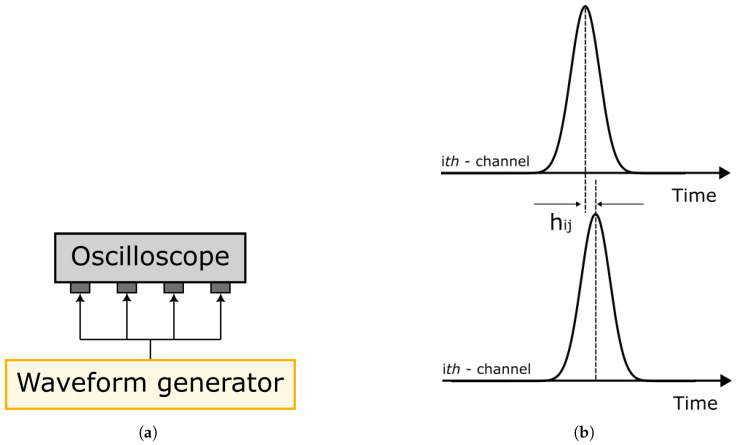
Jitter analysis setup. (**a**) Schematic of hardware measurement system. (**b**) Graphical definition of jitter analysis.

**Figure 3 sensors-23-01129-f003:**
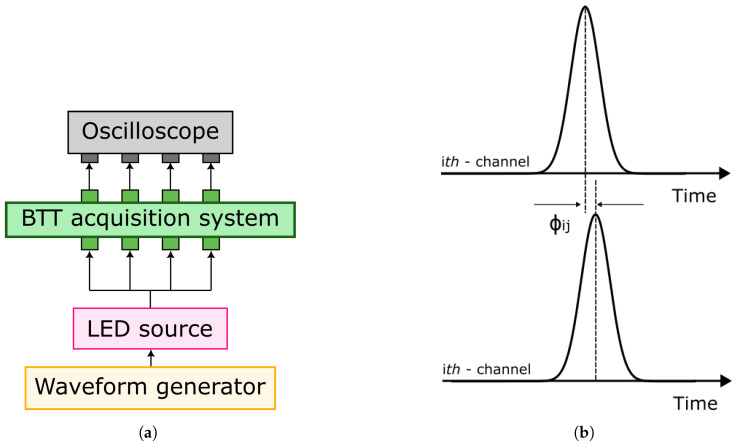
Probe setup. (**a**) Schematic of hardware measurement system. (**b**) Graphical representation of time-shift analysis.

**Figure 4 sensors-23-01129-f004:**
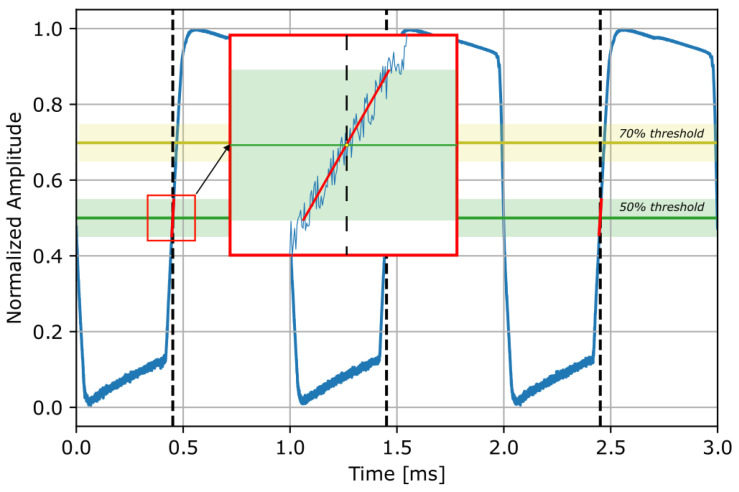
Threshold and interpolation process: example with 1 kHz wave frequency and 50% threshold level.

**Figure 5 sensors-23-01129-f005:**
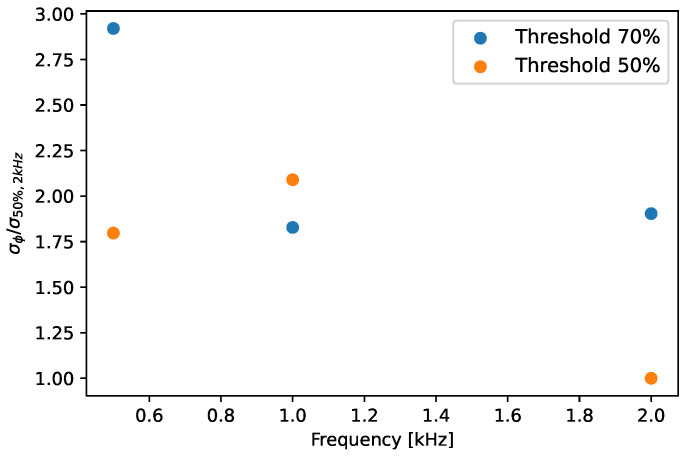
Time shift analysis: uncertainty introduced by the acquisition system due to wave frequency content and threshold level.

**Figure 6 sensors-23-01129-f006:**
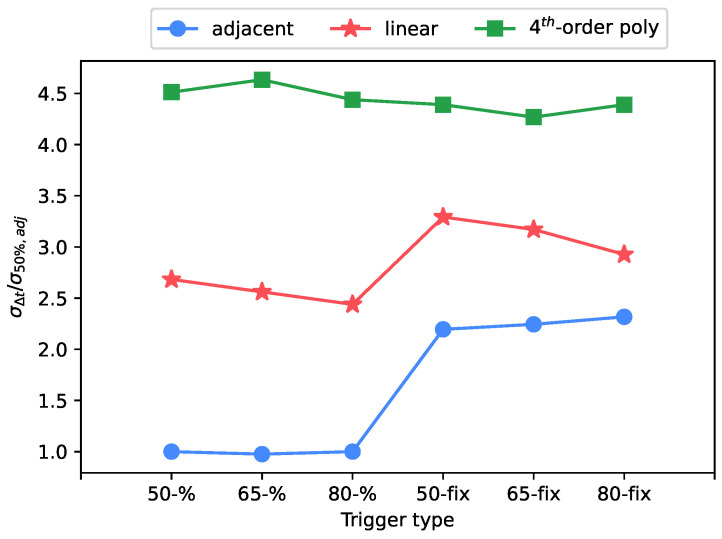
Standard deviation of arrival time shift: test case at 8.1 μs rise time, 0.5 V of full scale level.

**Figure 7 sensors-23-01129-f007:**
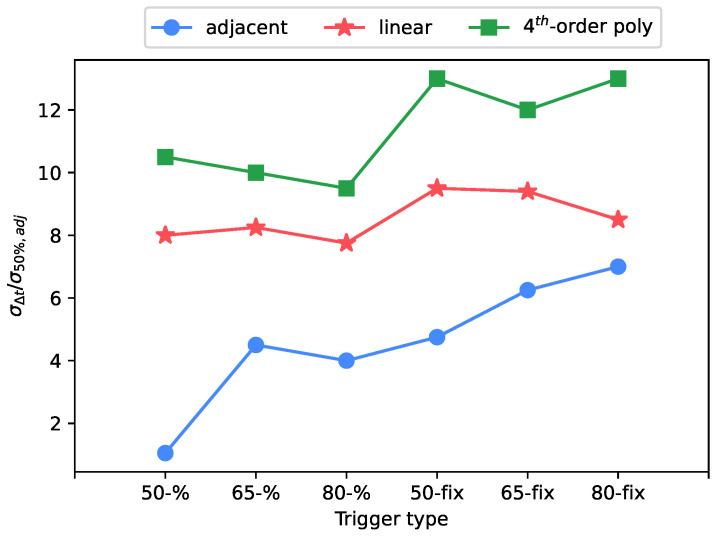
Standard deviation of arrival time shift: test case at 8.1 μs rise time, 1.5 V of full scale level.

**Figure 8 sensors-23-01129-f008:**
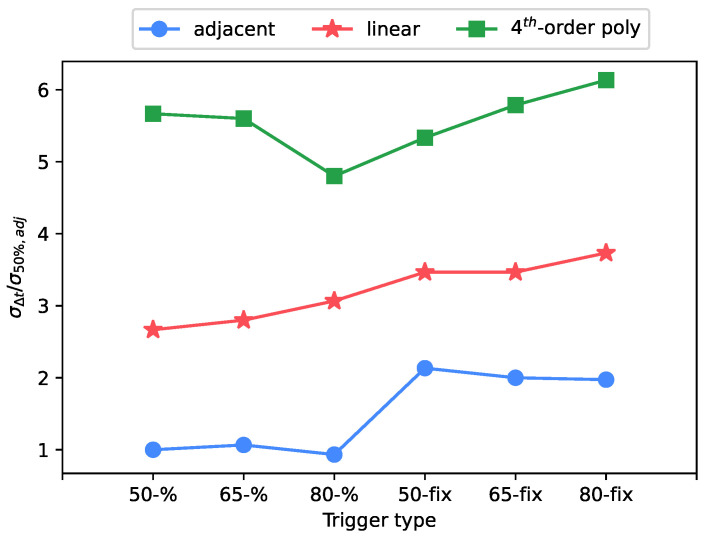
Standard deviation of arrival time shift: test case at 8.1 μs rise time, 1 V of full scale level.

**Figure 9 sensors-23-01129-f009:**
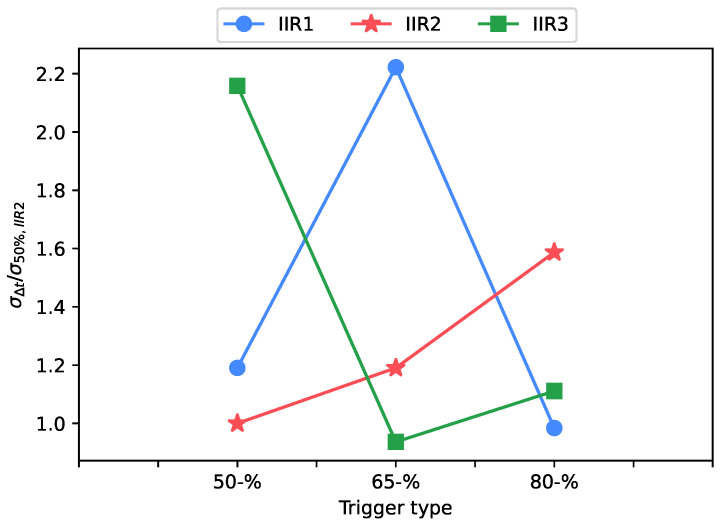
Standard deviation of arrival time shift: test case at 4 μs rise time, 1.5 V of full scale level.

**Table 1 sensors-23-01129-t001:** Rise time.

Wave Frequency (kHz)	Sampling Frequency (MHz)	Rise Time (μs)
20	1	8.1
40	1.5	4
60	2	2.7

**Table 2 sensors-23-01129-t002:** Set of parameters used in data processing analysis for a total of 342 combinations.

Rise Time (ms)	Interpolation Type	Full Scale (V)	Trigger Level/Type (V)	Edge
8.1	Adjacent	0.5	50%—percentage	Rising
			65%—percentage	
4	Linear fit	1	80%—percentage	
			50%—fixed	Falling
2.7	4th-order polynomial	1.5	65%—fixed	
			80%—fixed	

**Table 3 sensors-23-01129-t003:** Parameters used during the experimental tests with percentage trigger, adjacent interpolation and 1.5 V as full scale level.

Rise Time (ms)	Trigger Level/Type (V)	IIR Filter	Edge
8.1	50%—percentage	IIR1	
			Rising
4	65%—percentage	IIR2	
			Falling
2.7	80%—percentage	IIR3

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
