# Peer review of "Experimental Investigation on Hardware and Triggering Effect in Tip-Timing Measurement Uncertainty"

_sensors, 2023, doi:10.3390/s23031129_

Round 1

Reviewer 1 Report

The uncertainty sources on the arrval time estimation in a blade tip timing system are studied in this paper. The research sounds good, and the findings are interesting. Overall, the paper is well organized.

1. How is the uncertainty defined and indicated for the time of arrival? 2. What is the difference between Fai(ij) and hij? They are expected to be illustrated in a figure. 3. The variation trends of the standard deviation along the frequency are different for the threshold at 50% and 70% in Figure 5, explanation is expected.

Reviewer 2 Report

1. Write brief discussion on structural health monitoring.

2. What are the different operating conditions?

3. Provide more points on Figure 6.

4. At row number 76 check the spelling betewen. It should be between.

5. Authors mentioned that "In this way, a  total of 324 combinations of setting parameters are considered".  (Refer row no. 146). A detail analysis is needed.

6. Well documented conclusion.

Reviewer 3 Report

The title of this manuscript is very well written and practical, but unfortunately, the quality of the work done is not comparable to the title, and the authors need to do more detailed studies to be able to cover this title. In my opinion, this manuscript needs major editing in order to be published in this journal. However, below are some comments that authors should pay special attention to them:

1- Literature review should be improved and write about the work, used method, and results. More details should be explained and recently published papers should be used to do it. 

2- What is the novelty of the present work compared to other published papers. It should be bolded in the end of introduction section. 

3- Delete the last paragraph of the introduction section. 

4-  In section 2 and first line, it is strongly suggested to write about the name of non-contact sensor and its features. 

5- I can not see the results stated in this paper and extracted from Figure 5. please describe this subject exactly. for example, lines 107-108. 

6- It is necessary to describe physically reasons for all results and interpret the results. 

Round 2

Reviewer 3 Report

I am glad to inform you that I received the revision manuscript with marked changes and thank you for your kind mail.

I reviewed the revision manuscript and I also checked the authors’ responses to the reviewers comments.

In summary, I believe that the authors made their best efforts to edit the manuscript based on the reviewers comments and do it well. In addition, they response to all comments one by one. Now, I am agree to publish this manuscript in the present form.

Finally, it is Accepted.